# Videolaryngoendoscopic and Stroboscopic Evaluation in Predicting the Malignancy Risk of Vocal Fold Leukoplakia

**DOI:** 10.3390/jcm11195789

**Published:** 2022-09-29

**Authors:** Agata Leduchowska, Joanna Morawska, Wioletta Pietruszewska

**Affiliations:** Department of Otolaryngology, Head and Neck Oncology, Medical University of Lodz, 22 Kopcińskiego St., 90-153 Lodz, Poland

**Keywords:** leukoplakia, vocal fold, videolaryngoendoscopy, stroboscopy, laryngeal cancer

## Abstract

Background: Vocal fold leukoplakia (VFL), despite our knowledge of its etiopathogenetic factors, and the development of laryngeal visualization, remains a diagnostic and therapeutic challenge. Objective: This research aimed to explore the efficacy of clinical and morphological feature identification in videolaryngoendoscopy (VLE) using a three-tier classification, and videolaryngostroboscopy (VLS) in predicting the risk of VFL malignant transformation. Material and Methods: We examined 98 patients with VFL by flexible endoscopy under VLE and VLS. Morphological characteristics of 123 lesions including the surface, margin, and texture were assessed; then, VFL was subdivided into three types: I—flat and smooth, II—elevated and smooth, and III—rough. Based on the histopathological findings, 76 (61.79%) lesions were classified as low- and 47 (38.21%) lesions as high-grade dysplasia. Results: The inter-rater agreement between two raters evaluating the VFL in VLE was almost perfect (Cohen’s kappa = 0.826; *p* < 0.00; 95%CI 0.748–0.904). In ROC curve analysis, the AUC difference between Rater I and Rater II was 0.024 (0.726 vs. 0.702). In multivariate analysis, high-risk VFL was positively related to unilateral plaque localization (*p* = 0.003), the type III VLE classification (*p* = 0.013), absence of a mucosal wave (*p* = 0.034), and a positive history of alcohol consumption (*p* = 0.047). In ROC analysis, VLE had an AUC of 0.726, with a high sensitivity of 95.7% and low specificity of 40.8%. The NPV was high, at 93.9%; however, the PPV was low, at 50%. The proposed logistic regression model including features significant in multivariate analysis showed lower sensitivity (80.9% vs. 95.7%) and lower NPV (86.2% vs. 93.9%); however, the specificity and PPV were improved (73.7% vs. 40.8% and 65.5% vs. 50.0%, respectively). Conclusions: The combination of clinical history with endoscopic (plaque morphology) and stroboscopic examination (mucosal wave assessment) can fairly estimate the degree of dysplasia in VFL and thus is recommended for use in clinical settings. The findings of this study can be used to guide the decision regarding immediate biopsy or watchful waiting.

## 1. Introduction

One of the most common precancerous lesions in otolaryngological practice is vocal fold leukoplakia (VFL). The term “vocal fold leukoplakia” refers to thick whitish or grey patches without regard to their histological features and without implying any histological or prognostic information [1,2,3]. The pathologic results of clinically diagnosed VFL include hyperplasia, mild or moderate dysplasia, severe dysplasia, or carcinoma [4]. In accordance with the WHO (2017), laryngeal dysplasia is divided into low- and high-grade dysplasia [5].

The lesions can be either exophytic or flat, depending on the thickness of the keratin layer of the epithelium [6]. As the epithelium of the vocal folds is nonkeratinizing, leukoplakic change represents the alteration of the epithelium itself [7]. The terminology and classification of this spectrum of lesions have changed markedly over time. While in early studies, the term “keratosis” was used as a clinical term interchangeably with leukoplakia, in current terminology, “keratosis” is strictly reserved for a histological finding of a keratin layer on the squamous epithelium [8].

The average age of diagnosis of VFL is 50 years. The incidence reported in the USA is generally higher in males than in females, with 10.2 and 2.1 lesions per 100,000, respectively [9,10]. The current annual incidence of laryngeal dysplasia in Europe is not known [11].

The underlying etiologies for the development of leukoplakia patches or plaques are long-term smoking and alcohol abuse, inhaled irritant substances, and viral infections such as HPV [12]. Other possible etiological factors are occupational hazards, nutritional deficiencies, vocal misuse/abuse, chronic infections, hormonal disorders, and laryngopharyngeal reflux disease [8,13,14]. The role of infectious agents such as the human papillomavirus (HPV) is still a matter of debate [15]. Patients with VFL usually display voice issues similar to other vocal fold lesions, with hoarseness of the voice being the most common symptom and often the reason for a referral to an ENT doctor.

VFL has a malignancy transformation rate of approximately 8% [16]. Around 50% of patients clinically diagnosed with vocal cord leukoplakia have no dysplasia found in the histopathological examination; however, some of these lesions will eventually undergo malignant transformation [17]. A challenge in clinical practice is making a decision about which cases of leukoplakia indicate microlaryngoscopy with the histological examination [18]. Restoring voice capabilities and completely removing the lesion are the treatment goals in most cases [6]. For cases of VFL with a high possibility of malignancy, eradicating the cancer cells at an early stage is the most important goal, and deep resection may be necessary to achieve that, whereas a conservative treatment or watchful waiting policy benefits those with a low risk of malignancy [6].

Since survival significantly depends on the stage of detection of disease, it is important to focus early on the pathogenesis of laryngeal carcinoma at the precancerous or leukoplakic stage, to begin suitable therapy. Clinical diagnosis is made by imaging the larynx, and subsequently, by sampling the tissue, usually in the operating room. Histopathology has traditionally been used and is essential to determine the presence and degree of dysplasia in tissue samples [19].

The last two decades have seen major technological advancement in endoscopic methods, which has resulted in improved resolutions of images, contrast enhancement, magnification, and special filters for better exposure of the vascular structure of the mucosa. These high-quality digital systems, distal chips, and a filtered spectrum of light have improved the clinical accuracy in distinguishing between benign and malignant lesions. Videolaryngoendoscopy (VLE) remains the most widely used method for laryngeal examination. It relies on an analysis of a lesion’s visible morphological features. Specific morphological characteristics can vary and can be subtle, particularly in the early stages of the disease. As a result, any suspicious lesion is often biopsied for a definitive histopathological diagnosis. It should be noted that many of the characteristics are nonspecific and can be mimicked by benign pathologies, resulting in a significant number of unnecessary biopsies. Hence, several visualization techniques and technologies have been put forward to improve the diagnostic accuracy of physical examination of suspicious lesions, including VFL [20]. It is becoming common practice to supplement the VLE examination with videolaryngostroboscopy (VLS). This method is especially recommended for the evaluation of glottic lesions because of the possibility of assessing the vocal fold’s vibratory function. In a recent study, in 2021, Heyduck et al., suggested that even though the differentiation between moderate and mild dysplasia remains unsure, the risk of a malign transformation can be avoided with regular VLS examinations [19].

Along with new visualization techniques, in recent years, several studies have reported laryngeal leukoplakia classification using morphological characteristics [2,6,21,22,23].

The presented research aimed to predict the degree of VFL and assess the risk of its malignant transformation using clinical features, videolaryngoendoscopy, and videolaryngostroboscopy. The efficacy of this newest classification method, based on the assessment of morphological features, supplemented with stroboscopy, had not previously been studied extensively.

## 2. Materials and Methods

### 2.1. Patients

The study included 98 patients, comprising 19 women (19.39%) and 79 men (80.61%) aged from 38 to 85 (mean age 62.18), with clinically diagnosed vocal fold leukoplakia. Among the sample, 73 patients presented with unilateral VFL and 25 patients with bilateral VFL. In total, 123 cases of VFL were diagnosed. The patients were examined at the Otolaryngology, Head and Neck Oncology Department of the Medical University of Lodz from April 2015 to September 2020. The inclusion criteria were diagnosed VFL, no prior vocal fold-related medical intervention or procedure (surgery, radiation), and preoperative endoscopic assessment by VLE and VLS. The exclusion criterion was the disqualification of the patient from direct laryngoscopy with the collection of material for histopathological examination, e.g., when conservative treatment or work with a speech therapist could reduce pathological changes in the larynx. Furthermore, patients who had previously had surgical interventions with the vocal fold or were diagnosed with leukoplakia in a location other than the glottis were not included in the study. Patients who could not be operated on due to severe systemic diseases were excluded, as well. Approval for this study was granted by the Ethical Committee of the Medical University of Lodz (decision no. RNN/225/19/KE, 9 April 2019) and informed consent was obtained from each patient before their inclusion.

### 2.2. Methods

The medical history was gathered for each patient including their age, gender, lifestyle habits (alcohol use, smoking), and comorbidities. The endoscopic examination was performed using a transnasal flexible video endoscope with an LED light (model CV-170 HD, ENF-VH, Olympus Corp., Tokyo, Japan) by an experienced team of otorhinolaryngologists. After VLE examination, the endoscope mode was switched to VLS. The patients were examined while in the sitting position after the application of nasal cavity anesthesia. Topically applied lidocaine spray was administered in the throat if necessary (in the case of patients with strong gag reflex). The endoscope was inserted through the wider nasal cavity.

Morphological classification of VFL was performed according to the protocol proposed by Chen et al. (2019). Two laryngologists (AL, WP) blinded to the patients’ identities and pathological results concurrently but independently assessed the lesions. Morphological characteristics including the surface, margin, and texture were recorded; then, vocal fold leukoplakia was subdivided into three types: I—flat and smooth, II—elevated and smooth, and III—rough (Figure 1). Types I and II were considered low-risk and type III high-risk lesions. Localization of the lesions (unilateral/bilateral), focality (unifocal/multifocal), anterior commissure involvement, inferior vocal fold edge involvement, and the presence of a mucosal wave in the VLS were determined.

The patients underwent phonosurgical treatment of their leukoplastic lesions to remove the lesions, and we verified the findings of the endoscopic assessment using a histopathological examination. All patients underwent surgical removal of the lesions under general anesthesia by endotracheal intubation. The final diagnosis was established based on histopathological examinations of the specimens. The morphology of vocal fold leukoplakia was compared with the respective histopathological (HP) diagnosis. Leukoplakia was divided into two types: (1) low-grade dysplasia (squamous hyperplasia or mild dysplasia); (2) high-grade dysplasia (moderate dysplasia, severe dysplasia, carcinoma in situ, or invasive carcinoma) [5].

### 2.3. Statistical Analysis

Data were analyzed using the STATISTICA 13.1 software (Dell, Round Rock, TX, USA). Continuous variables were tested for normality using the Shapiro–Wilk test. To compare independent groups, a Kruskal–Wallis non-parametric ANOVA was conducted with Dunn’s post hoc test. In all statistical tests, a two-tailed *p*-value of 0.05 was considered statistically significant.

The sensitivity, specificity, and positive and negative predictive values were calculated, and an optimal cut-off value was established based on the receiver operating curve (ROC) analysis [24].

Cohen’s kappa index (κ) was used to assess the inter-rater agreement between two physicians evaluating the endoscopic images, along with the intra-rater agreement for both Rater I and Rater II. We used Landis and Koch’s commonly applied guidelines to interpret Cohen’s κ: (κ) 0.00–0.20 indicates slight agreement, (κ) 0.21–0.40 fair agreement, (κ) 0.41–0.60 moderate agreement, (κ) 0.61–0.80 substantial agreement, and (κ) 0.81–1.00 almost perfect agreement [25].

The univariate and multivariate analysis was performed using logistic regression, the results of which are presented in a format suggested by Peng et al. [26]. To perform logistic regression analysis on the probability of high-risk or low-risk leukoplakia, the following leukoplakia and epidemiological characteristics were analyzed in univariate analysis: age, gender, leukoplakia localization, focality, involvement of the anterior commissure, involvement of the inferior margin of the vocal fold, history of smoking, alcohol consumption, VLE classification, and presence of a mucosal wave in the VLS. All variables were analyzed using the likelihood ratio (LR) test. The test results were considered statistically significant at a *p*-value < 0.05.

The variables were checked for interactions and linearity of predictors using the LR test. No interactions between variables were detected. The linearity test result for age was *p* = 0.025; therefore, the variable was non-linear. Age was then categorized into dichotomic categories and included in further analysis. A mixed-effects logistic regression model was constructed to model a binary outcome of 0: low-risk leukoplakia, 1: high-risk leukoplakia. The predictive ability of the model was validated using the V-fold cross-validation method with 10 subsets of data. The goodness-of-fit of the model was evaluated with the Hosmer–Lemeshow test.

## 3. Results

### 3.1. Patients

There were 123 cases of VF leukoplakia from 98 examined patients who met the inclusion and exclusion criteria. In terms of localization of the lesions, unilateral vocal fold leukoplakia was diagnosed in 73 patients (74.49%) and bilateral vocal fold leukoplakia in the remaining 25 patients (25.51%). Based on the histopathological findings 76 (61.79%) lesions were classified as low-grade dysplasia and 47 (38.21%) lesions as high-grade dysplasia. Demographic and clinical data are presented in Table 1.

### 3.2. Statistical Analysis

#### 3.2.1. Inter-Rater and Intra-Rater Agreement

The inter-rater agreement between two raters evaluating the VFL in white light endoscopy according to Chen’s classification from 2019 was almost perfect, with Cohen’s kappa = 0.826 (*p* < 0.001), 95%CI (0.748; 0.904). In ROC curve analysis for both observers, the AUC difference between Rater I and Rater II was 0.024 (0.726 vs. 0.702); therefore, in further results, we only report ROC curve parameters from Rater I.

In the results from both raters (Table 2), significant differences in pathological grades were witnessed among some morphological types according to a Kruskal–Wallis ANOVA test followed by Dunn’s test (flat and smooth vs. elevated and smooth, *p*_a_ = 0.001 and *p*_a_ = 0.004; flat and smooth vs. rough, *p*_b_ = 0.001 and *p*_b_ = 0.002), but for the comparison of the morphological types elevated and smooth vs. rough, the results were statistically non-significant (*p* = 1.0).

The VLE classification according to Chen (2019) and mucosal wave in the VLS were evaluated twice, once for an initial evaluation, then subsequently during anonymized video playback. The results are presented in Table 3.

#### 3.2.2. Univariate Analysis

Univariate analysis revealed that unilateral vocal fold changes were 4.11 times more likely to present as high-risk VFL (*p* < 0.001), and a history of alcohol consumption (*p* = 0.010; OR 3.071) was associated with a three-fold increase in the odds of developing a malignancy (Table 4). These observations were even stronger after conducting a multivariate analysis, which is described further.

The odds of developing high-risk VFL increased by 0.04 times with every year of age. However, since age was found to be a non-linear predictor, we categorized the age of patients to below 60 years and older than 60, which resulted in a stronger observation. In this case, subjects/patients above 60 were 2.38 times more likely to develop a malignant transformation under the leukoplakia plaque. A history of smoking (current or former tobacco users) was associated with a 5.081 times greater increase in the probability of developing high-risk VFL.

The absence of a mucosal wave in a VLS examination can be considered a risk factor, with an 8.75 (*p* < 0.001)-fold increase in the odds of malignant transformation.

An important predictor in VLE analysis could be the involvement of the inferior surface of the vocal fold (*p* = 0.002; OR 3.321); however, these observations were not found to improve the overall predictive capabilities of the multivariate logistic regression model.

The type III VLE classification was most strongly associated with an increased risk of cancer within leukoplakia plaque (OR 19.207; *p* < 0.001).

The gender, anterior commissure involvement, and focality of leukoplakia (multifocal or unifocal plaques on one vocal fold) were not associated with a statistically significant greater chance of malignant transformation within VFL.

#### 3.2.3. Multivariate Analysis

According to the model, the log of the odds of high-risk leukoplakia was positively related to unilateral plaque localization (*p* = 0.003), the type III VLE classification (*p* = 0.013), absence of a mucosal wave (*p* = 0.034), and a positive history of alcohol consumption (*p* = 0.047). Moreover, the odds of malignant transformation developing within leukoplakia plaque in unilateral changes were 5.458 times greater than in bilateral VFL. This finding was confirmed by the positive coefficient (1.697) associated with the leukoplakia localization predictor.

The odds of high-risk VFL in the histopathological examination were 9.314 times higher when the leukoplakia plaque in VLE was not only elevated but also rough (Type III) compared to flat or elevated plaque but with a smooth surface (Type I and II). Confirmation of this observation may be derived from the positive regression coefficient of 2.231. Moreover, the absence of a mucosal wave is correlated with 4.479 times greater odds of developing high-risk VFL. Finally, considering the patient’s history, alcohol consumption is associated with a 3.357 times higher risk of developing a malignancy. The results of multivariate analysis are presented in Table 5.

To determine an individual’s predisposition to high-risk leukoplakia (the risk of developing laryngeal cancer within leukoplakia in an individual patient), a logistic regression equation was plotted taking into account characteristics with independent prognostic significance for the occurrence of high-grade leukoplakia (Figure 2).

The individual predisposition to high-risk leukoplakia (IPHRL) was determined by the formula: IPHRL = 1/(1 + e^−logit(p)^). For example, the IPHRL for a patient not consuming alcohol, with diagnosed unilateral Type III leukoplakia in VLE, without a preserved mucosal wave, is 50.42%. Alternatively, the presence of all features with a significant influence increases the risk of developing high-risk leukoplakia to 77.35%.

#### 3.2.4. Diagnostic Performance

The results of our VLE classification and histopathological examination were compared. To evaluate the diagnostic performance of the VLE classification regarding the risk of malignant transformation in clinical leukoplakia we compared the AUC values in a ROC curve analysis, along with the sensitivity, specificity, and positive and negative predictive values. In the ROC analysis, VLE evaluation had an AUC of 0.726, with a high sensitivity of 95.7%, low specificity of 40.8%, and high NPV of 93.9%. However, the PPV was low, at 50%. The results are summarized in Table 6.

The proposed logistic regression model, which includes evaluation of the patient in terms of VLE, history of alcohol consumption, mucosal wave presence, and VFL localization (uni- or bilaterality), improves the clinical utility of VLE morphological classification and shows improved diagnostic performance. The AUC of this model is higher (0.861). This model shows a lower sensitivity (80.9% vs. 95.7%) and NPV (86.2% vs. 93.9%), but the specificity and PPV are improved (73.7% vs. 40.8% and 65.5% vs. 50.0%, respectively).

## 4. Discussion

Vocal cord leukoplakia is a clinical diagnosis regarded as a precancerous condition [27]. One of the challenges in managing VFL is to determine the potential for malignant transformation of the benign and premalignant lesions, and thus to properly assess the need for surgical intervention. The histology and behavior of glottic keratosis can be frustratingly difficult to predict as lesions can present on a spectrum anywhere from completely benign growth to invasive malignancy [28]. Studies have reported that the clinical diagnosis of leukoplakia represents an approximately 6–7% chance of progressing into carcinoma [17,29]. A macroscopic appearance and its correlation with malignancy are of special interest to clinicians due to the potential for guiding decision-making at the time of the initial laryngoscopic examination [28]. However, even when a leukoplakia does not show a positive histopathological result for dysplasia, there is a 3.6–30% risk that it could develop into an invasive carcinoma [19].

VLE plays the main role in the diagnosis of patients with laryngeal diseases and, to date, is the most widely used diagnostic tool in the assessment of premalignant lesions or cancer of the larynx [30]. Beyond that, VLS, which shows the mucosal wave, provides additional information on the depth of the vocal fold involvement according to pathological changes. The key goal of the endoscopic examination is to decide whether or not a lesion is malignant. When the result of the examination is inconclusive, surgical microlaryngoscopy is often performed to obtain a histopathological diagnosis. Even though the result might be benign, this creates a risk of surgical complications because the deeper the resection in the vocal fold, glottic surface, and anterior commissure, the greater the scarring, theoretical vocal impact, and consequent dysphonia [15]. On the other hand, a lesion should not be left under observation if malignant transformation could begin. Therefore, attempts are being made to determine the features differentiating low- and high-risk leukoplakia at the stage of the clinical examination, taking into account data such as the age, gender, smoking history, and alcohol consumption, supplemented by endoscopic and stroboscopic examination findings. In VLE, the mucosal appearance of the vocal leukoplakia can be classified using multiple factors, such as the homogeneity of color and regularity of texture. Recording these findings in a standard form is helpful for medical communication and follow-up. To that end, it is important to establish a scoring system that is reliable and also has clinical significance [6].

There are several reports on the clinical classification of vocal fold leukoplakia. Fang et al. (2016) aimed to establish a scoring system by combining clinical demographic and laryngoscopic characteristics, to improve the management of VFL. They classified lesions based on their morphology, including the color, texture, size, hyperemia, thickness, and symmetry [21]. Li et al., meanwhile, classified VFL lesions into superficial, exophytic, and ulcerative lesions [2]. In a further study, Chen et al., divided lesions into three groups: flat and smooth, elevated and smooth, and rough [31]. In these studies, lesions were classified as low- and high-grade and the treatment decision was made according to this classification [32].

Yet, there are some limitations to the described classifications. While the classification by Fang et al., is very detailed, it may be difficult to incorporate this form of VFL assessment into daily clinical practice as it requires detailed analysis of six particular aspects of the VFL morphology. Thus, the assessment is time-consuming and should be performed by a specialist with substantial experience. The classification by Lee et al., in contrast, is short and easy to perform in daily clinical practice, but as the authors admit, it is based only on the texture of the lesions, and it mainly reflects the evolutional process of vocal cord leukoplakia from early- to late-stage [2].

In 2019, a new classification was proposed by Chen et al., based on morphological characteristics of leukoplakia. At the time when we set out to conduct this study, there has been no research exploring its efficacy. The authors of the present study examined 98 patients with 123 vocal fold lesions to evaluate the diagnostic value of the Chen (2019) classification. We used the correlation between the results of the morphological assessment and histological diagnosis to assess the clinical applicability of the newly proposed classification for everyday clinical practice. We have demonstrated its usefulness in daily clinical practice and identified type III patients as significantly at risk of cancer with the VFL. VLE was used to assess the morphological structure of leukoplakia based on the proposed three-tier classification. The authors tried to determine to what extent each of the three morphological aspects, namely the surface, margin, and texture of the lesion, should be considered to accurately determine the presence of malignancy or lack thereof.

Although VLE plays an essential role in the diagnosis of vocal cord diseases, even for experienced doctors, it is difficult to distinguish benign lesions from malignant lesions [33]. Nonetheless, in our study, the percentage inter-observer agreement of the morphological classification was almost perfect (kappa = 0.826, *p* < 0.001). In the original study by Chen et al. [23], that agreement was slightly lower, at 78.7% (kappa = 0.615, *p* < 0.001). Our improved agreement confirms that the examined classification is, as the authors intended, both simple and comprehensive.

Ricci and Isenberg reported that approximately 50% of patients with a clinical diagnosis of vocal fold leukoplakia do not have dysplasia, making a point that these patients receive unnecessary surgical treatment [17]. Our research confirms that classifying the lesions as benign allows time for conservative treatment and adopting a watchful waiting policy (Types I and II according to Chen (2019)), while classifying lesions as suspicious (Type III) indicates that surgical treatment with histopathological verification is required and should not be delayed. It is worth noting, however, that if the “watchful waiting” approach is routinely recommended, patients with dysplasia potentially advancing into cancers may be missed. [16] In this sense, adopting this classification might best serve as a good indicator when prioritizing which patients should undergo microsurgical biopsy immediately.

In the original study by Chen et al. [23], the authors admitted that they did not investigate the specific predictive factors for pathological grades of lesions, such as tobacco smoking, alcohol intake, and laryngopharyngeal reflux [23]. However, such reliable prognostic factors have important clinical implications [3].

In our study, several factors were taken into account (Table 4) but the clinical utility of the classification was improved by only three of them, namely the history of alcohol consumption, mucosal wave presence, and VFL localization (uni- or bilaterality). It has indeed been demonstrated in epidemiological studies that alcohol consumption is an independent risk factor for the transformation of vocal fold leukoplakia to laryngeal cancer, and the risk increases with the amount of alcohol consumed [34]. Regarding the absence of a mucosal wave, meanwhile, the analysis of the clinical parameters of VF leukoplakia performed by Cui et al., indicated it to be highly associated (*p* < 0.001) with the histopathological results [29].

Although cigarette smoking was entered into multivariable modeling, it was ultimately removed due to the lack of significance in this dataset. This obviously does not exclude smoking as a risk factor or undermine its importance. It was likely not significant because 87% of the studied group were smokers, making the non-smoking group too small to provide a suitably good reference with enough statistical power for smoking to reach significance.

Even though a vast number of publications have confirmed noninvasive biological endoscopy, such as narrow-band imaging (NBI) and Storz SPIES [30,35,36,37], to have great accuracy when predicting the final histopathology results for glottic lesions, a gold-standard policy for the management and follow-up of precancerous laryngeal lesions has not yet been defined [38]. The NBI technique, for instance, has always, with few exceptions, been assessed alongside VLE [39].

The above-mentioned methods are very useful tools, but they require additional, cost-incurring equipment and the learning curve is relatively long [40], while VLE supplemented with VLS remains the key clinical element for detecting and assessing vocal fold lesions [19,21].

It should be underlined that in a great number of institutions, VLE is still the only method at clinicians’ disposal. Hence, the application of the proposed scoring system for VLE could help ensure timely detection of leukoplakia lesions with malignant transformation potential and thereby prevent unnecessary surgery. Given that a histological classification can only be made after excising lesions, and severe hoarseness may occur post-surgery, it is important to identify preoperative factors and classify lesions before excision, and in this way, decrease the incidence of such complications [21].

The research presented here had some limitations. Our goal was to present a ready-to-use, effective tool in daily clinical practice. In the future, this could be extended with biological endoscopy methods; however, it should be kept in mind that they require a long learning curve and equipment availability.

## 5. Conclusions

Vocal fold leukoplakia can be classified according to the plaque morphology, distinguishing three types: flat and smooth, elevated and smooth, and rough. This classification, despite being a subjective method of assessment, seems to be consistent between independent raters.

Morphological types of VFL correlate with the degree of histopathological advancement—type III according to Chen (2019) is the predictor of malignant transformation. For institutions that do not have NBI at their disposal, the analyzed classification is a helpful diagnostic tool for assessing precancerous lesions as it allows for accurate detection of benign lesions. Localization of the lesions to one vocal fold, absence of a mucosal wave, an alcohol consumption history, and the morphological type of the lesion may increase the possibility of malignant transformation. Armed with this knowledge, clinicians can plan treatments accordingly.

Videolaryngoendoscopy, complemented by videolaryngostroboscopy, which has previously been used mainly for the functional assessment of the vocal folds, is now becoming an indispensable tool in everyday clinical practice for a head and neck oncologist. Beyond the morphology of the plaque, the presence of a mucosal wave is also crucial for complex prognosing of high-risk vocal fold leukoplakia.

## Figures and Tables

**Figure 1 jcm-11-05789-f001:**
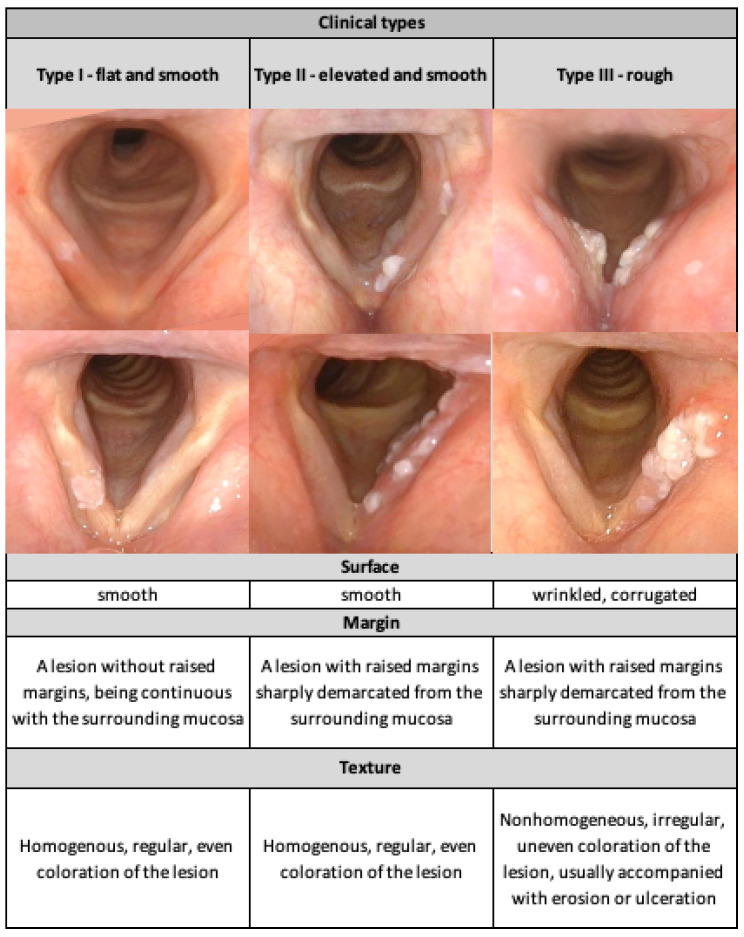
Morphological types of leukoplakia according to Chen (2019) [23], based on the authors’ repository images from patients with VFL (N = 98) who were enrolled in the study.

**Figure 2 jcm-11-05789-f002:**
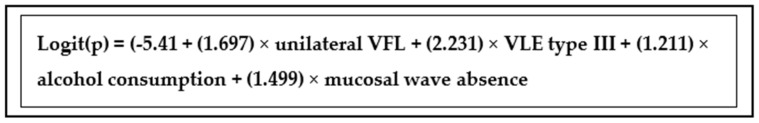
Equation for predicting the probability of developing high-risk leukoplakia (for all samples, if a given feature is present, then it is substituted with 1 in the equation, or if it is absent, then it is substituted with 0).

**Table 1 jcm-11-05789-t001:** Clinical characteristics of vocal fold leukoplakia in the enrolled subjects.

VLE Classification		Type I: Flat and Smooth N (%)	Type II: Elevated and Smooth N (%)	Type III: Elevated and Rough N (%)	Total N (%)
**Number**		37 (30.08)	60 (48.78%)	26 (21.14)	123 (100)
**Gender**	Male	26 (26.80)	51 (52.58%)	20 (20.62)	97 (78.86)
	Female	11 (42.31)	9 (34.62%)	6 (23.08)	26 (21.14)
**Age**	Mean (range; SD)	56.08 (38–76; 11.409)	64.55 (45–85; 8.852)	65.38 (55–80; 6.777)	62.18 (38–85; 10.097)
**Smoking**	No	9 (56.25)	5 (31.25)	2 (12.50)	16 (13.00)
	Yes	28 (26.17)	55 (51.40)	24 (22.43)	107 (87.00)
**Alcohol consumption**	No	15 (36.59)	20 (48.78)	6 (14.63)	41 (33.33)
	Yes	22 (26.83)	40 (48.78)	20 (24.39)	82 (66.67)
**Localization**	Unilateral	16 (21.92)	36 (49.32)	21 (28.77)	73 (59.35)
	Bilateral	21 (42.00)	24 (48.00)	5 (10.00)	50 (40.65)
**Focality**	Unifocal	26 (35.62)	36 (49.32)	11 (15.07)	73 (59.35)
	Multifocal	11 (22.00)	24 (48.00)	15 (30.00)	50 (40.65)
**Anterior commissure involvement**	No	30 (41.10)	33 (45.21)	10 (13.69)	73 (59.35)
	Yes	7 (14.00)	27 (54.00)	16 (32.00)	50 (40.65)
**Inferior edge involvement**	No	27 (42.19)	32 (50.00)	5 (7.81)	64 (52.03)
	Yes	10 (16.95)	28 (47.46)	21 (35.59)	59 (47.97)
**Mucosal wave presence**	No	1 (1.85)	34 (62.96)	19 (35.19)	54 (43.90)
	Yes	36 (52.17)	26 (37.68)	7 (10.14)	69 (56.10)

**Table 2 jcm-11-05789-t002:** Videolaryngoendoscopy classification of vocal fold leukoplakia according to Rater I and Rater II.

Number of Cases	Type I—Flat and Smooth		Type II—Elevated and Smooth		Type III—Rough		*p*-Value	
	Rater I	Rater II	Rater I	Rater II	Rater I	Rater II	Rater I	Rater II
High-risk VFL	2	2	30	25	15	20	*p* < 0.001 * *p*_a_ = 0.001 *p*_b_ = 0.001 *p*_c_ = 1.0	*p* = 0.018 * *p*_a_ = 0.004 *p*_b_ = 0.002 *p*_c_ = 1.0
Low-risk VFL	35	31	30	28	11	17		
**Inter-rater agreement (Rater I vs. Rater II)**							Cohen’s kappa = 0.826 *p* < 0.001 **	

* Kruskal–Wallis ANOVA; ** Bowker’s test for symmetry; *p*_a_ Kruskal–Wallis ANOVA with Dunn’s post hoc test, flat and smooth vs. elevated and smooth; *p*_b_ Kruskal–Wallis ANOVA with Dunn’s post hoc test, flat and smooth vs. rough; *p*_c_ Kruskal–Wallis ANOVA with Dunn’s post hoc test, elevated and smooth vs. rough.

**Table 3 jcm-11-05789-t003:** The inter-rater agreement (between Rater I and Rater II) and intra-rater agreement in the evaluation of: the leukoplakia plaque morphology according to Chen (2019), and a mucosal wave presence on the vocal fold.

	Cohen’s Kappa
**VLE classification**	
Inter-rater (Rater I—Rater II)	0.826 (*p* < 0.001) 95%CI (0.748; 0.904)
Intra-rater I	0.895 (*p* < 0.001) 95%CI (0.767; 1.024)
Intra-rater II	0.875 (*p* < 0.001) 95%CI (0.749; 1.001)
**Mucosal wave evaluation in VLS**	
Inter-rater (Rater I—Rater II)	0.785 (*p* < 0.001) 95%CI (0.608; 0.962)
Intra-rater I	0.867 (*p* < 0.001) 95%CI (0.690; 1.043)
Intra-rater II	0.816 (*p* < 0.001) 95%CI (0.640; 0.993)

**Table 4 jcm-11-05789-t004:** Summary of univariate analysis conducted as the first step toward formulating a logistic regression model. LR—likelihood ratio test; OR—odds ratio.

Variable	*p*-Value [LR Test]	Included in Multivariate Analysis	OR [95%CI]
Age	0.049	No	1.040 (1.000; 1.081)
Age 60 yr. and older	0.037	Yes	2.380 (1.054; 5.374)
Male gender	0.079	No	2.44 (0.900; 6.616)
Unilateral leukoplakia localization	<0.001	Yes	4.11 (1.791; 9.439)
Unifocality of leukoplakia	0.676	No	0.853 (0.406; 1.796)
Anterior commissure involvement	0.066	No	2.07 (0.955; 4.217)
Inferior vocal fold edge involvement	0.002	Yes	3.321 (1.550; 7.119)
Mucosal wave absence	<0.001	Yes	8.750 (3.791; 20.194)
Current or former smoking	0.037	Yes	5.081 (1.100; 23.476)
Alcohol consumption	0.010	Yes	3.071 (1.303; 7.238)
Type III in VLE classification	<0.001	Yes	19.207(4.344; 84.925)

**Table 5 jcm-11-05789-t005:** Summary of multivariate analysis using logistic regression, including the parameters for the regression model and the goodness-of-fit test results.

Predictor	β	Wald’s χ^2^	*p*	OR [95%CI]
**Constant**	−5.410	3.617	0.057	NA
**Unilateral leukoplakia localization**	1.697	8.595	0.003	5.458 (1.755–16.973)
**VLE classification—Type III**	2.231	6.112	0.013	9.314 (1.588–54.628)
**Mucosal wave absence**	1.499	4.510	0.034	4.479 (1.123–17.870)
**Alcohol consumption**	1.211	3.940	0.047	3.357 (1.015–11.101)
**Regression model statistics**				
**Parameter**	**Test outcome**			
Wald’s test	χ^2^ = 26.149 D*f = 10* *p = 0.004*			
Hosmer-Lemeshow test	5.485 *p* = 0.705			
AIC	128.357			

AIC—Akaike Information Criterion.

**Table 6 jcm-11-05789-t006:** Diagnostic value of VLE classification.

Clinical Classification	Proposed Cut-Off Value	AUC	Sensitivity	Specificity	PPV	NPV
**VLE classification**	Type III	0.726	95.7%	40.8%	50.0%	93.9%
**The proposed logistic regression model for IPHRL**	n/a	0.861	80.9%	73.7%	65.5%	86.2%

AUC—area under the curve, PPV—positive predictive value, NPV—negative predictive value, n/a—not available, IPHRL—individual predisposition to high-risk leukoplakia.

## Data Availability

Data available on request.

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
