# Peer review of "Videolaryngoendoscopic and Stroboscopic Evaluation in Predicting the Malignancy Risk of Vocal Fold Leukoplakia"

_jcm, 2022, doi:10.3390/jcm11195789_

Round 1
Reviewer 1 Report
This manuscript studied the efficacy of physical characteristics on vocal fold leucoplakia in white light (VLE) and VLS in diagnosing low grade and high grade dysplasia. Using multivariate logistic regression the potential of high grade dysplasia diagnosis is calculated. It is a rigorous statistical consideration that includes the unilaterality, alcohol consumption, type of VFL and absence of mucosal wave.
To improve the strength of the results of qualitative evaluation, other that inter-rater agreement, the authors must report the intra-rater agreement as well.
The equation that predicts the probability of high risk VF leucoplakia includes absence of mucosal waves. Thus, the authors must report the inter-rater and intra-rater reliability of evaluation of mucosal wave on VLS. What evaluation tool is used to analyse the mucosal wave? Typically, it is not just reported as absent or present. Its amplitude can be mild or severely reduced.
Cigarette smoking is a known risk factor for premalignant or malignant VF lesions. The authors must discuss the reasons why smoking is not shown to be a risk factor hence it was excluded from the equation. The conclusion of this study must be stated with caution and the limitations are also discussed.
Author Response
Reviewer 1
Dear Reviewer,
Thank you very much for your thoughtful comments and efforts toward improving our manuscript.
We would like to address your concerns and comments point by point:
- To improve the strength of the results of qualitative evaluation, other that inter-rater agreement, the authors must report the intra-rater agreement as well.
We have added the details on the intra-rater agreement for qualitative evaluation, that is mucosal wave assessment and plaque morphology assessment. The table presenting the intra-rater agreement has been added to the manuscript:
Table 3. The inter-rater agreement (between Rater I and Rater II) and intra-rater agreement in the evaluation of: leukoplakia plaque morphology acc. to Chen 2019; and mucosal wave presence on vocal fold.
|
Cohen’s kappa |
VLE classification |
|
Inter-rater (Rater I – Rater II) |
0.826 (p <0.001) 95%CI (0.748;0.904) |
Intra-rater I |
0.895 (p <0.001) 95%CI (0.767;1.024) |
Intra-rater II |
0.875 (p <0.001) 95%CI (0.749;1.001) |
Mucosal wave evaluation in VLS |
|
Inter-rater (Rater I – Rater II) |
0.785 (p <0.001) 95%CI (0.608;0.962) |
Intra-rater I |
0.867 (p <0.001) 95%CI (0.690;1.043) |
Intra-rater II |
0.816 (p <0.001) 95%CI (0.640;0.993) |
- The equation that predicts the probability of high-risk VF leukoplakia includes absence of mucosal waves. Thus, the authors must report the inter-rater and intra-rater reliability of evaluation of mucosal wave on VLS. What evaluation tool is used to analyse the mucosal wave? Typically, it is not just reported as absent or present. Its amplitude can be mild or severely reduced
The inter-rater and intra-rater reliability of evaluation of mucosal wave on VLS were provided, and added in the manuscript. The mucosal wave presence/absence was evaluated twice -for the first time upon initial evaluation, and subsequently during anonymized video playback. We based our evaluation on the Protocol for Instrumental Assessment of Vocal Function by Speech- Language-Hearing Association Expert Panel (Patel et al. 2018) which suggests the mucosal wave extent should be rated as the observation of mucosal wave movement from the medial edge toward the lateral surface of the vocal fold in increments of 25%, ranging from 0% to 100%, where 100% refers to the total visible width of the vocal fold. In our study, the range of 0-25% was classified as ABSENT and >25% as PRESENT.
Patel RR, Awan SN, Barkmeier-Kraemer J, Courey M, Deliyski D, Eadie T, Paul D, Švec JG, Hillman R. Recommended Protocols for Instrumental Assessment of Voice: American Speech-Language-Hearing Association Expert Panel to Develop a Protocol for Instrumental Assessment of Vocal Function. Am J Speech Lang Pathol. 2018 Aug 6;27(3):887-905. doi: 10.1044/2018_AJSLP-17-0009. PMID: 29955816.
- Cigarette smoking is a known risk factor for premalignant or malignant VF lesions. The authors must discuss the reasons why smoking is not shown to be a risk factor hence it was excluded from the equation. The conclusion of this study must be stated with caution and the limitations are also discussed.
Cigarette smoking was entered into multivariable modelling but was ultimately removed due to the lack of significance in this dataset. This obviously does not exclude smoking as a risk factor or underscore its importance. It is likely not significant because 87% of the studied group were smokers, making the non-smoking group too small to provide a suitably good reference with enough statistical power for smoking to reach significance. Clearly, non-smokers tended to show type I lesions while non-smokers mostly showed type II or type III (as evidenced in table 4 – p-value 0.037). Finally, given the significant overlap between smoking and drinking alcohol, it is likely that the collinearity of the two variables leads to the exclusion of one. Again, this does not rule out smoking as a risk factor, just shows that in this set of the specific sample size and distribution of risk factors alcohol consumption reach statistical significance while smoking did not, although the effect of both was similar in terms of magnitude and direction in univariate analysis. Table 5 shows results of multivariable regression where alcohol consumption remained in the model alongside type III lesions, which in itself was more prevalent among smokers. It is therefore likely that the overlap of smoking/alcohol consumption increased the likelihood of type III lesions and this factor “captured” most of the effect of smoking in terms of risk in the final classification model. Ultimately, the model was intended for clinical evaluation on the basis of accuracy. Its creation was thus driven by the significance of individual variables and error minimization at the same time striving to reach the minimal model capable of class separation (hence the AIC criterion given in table 5). This likely led to the model selecting variables on the basis of their contribution toward diagnostic performance and once alcohol consumption and type III lesions were included there was no benefit that could be achieved by the addition of smoking in a group where ~90% of samples were positive for this factor.
Reviewer 2 Report
The study tries to highlight the effectiveness of videolaryngoendoscopy (VLE) using a three-level classification and videolaryngostroboscopy (VLS) to establish some clinical and morphological characteristics that can predict the risk of malignant transformation in the case of leukoplasia of the vocal folds.
The study was conducted on 98 patients (19 women and 79 men) with clinically diagnosed vocal cord leukoplakia. The pathological results were evaluated simultaneously, but in an independent manner, by two laryngology specialists, which ensured a rigorous identification of patients with low-risk lesions compared to patients with high-risk lesions of malignant transformation.
For the data analysis, a dedicated software STATISTICA 13.1 (Dell, USA), was used, allowing of the comparison of independent groups, but also a multivariate analysis.
The presented results show that videolaryngoendoscopy, supplemented by videolaryngostroboscopy, previously used only for the functional evaluation of the vocal cords, is now becoming an indispensable tool in clinical practice.
Endoscopic examination (plaque morphology) and stroboscopic examination (mucous wave evaluation) together with the clinical history of each patient can help to correctly determine the degree of dysplasia in the VFL and suggest the appropriate clinical approach.
The conclusions are supported by the data presented, this paper being of interest considering the original approach on the very hot topic addressed.
The manuscript is based on thorough research of the scientific literature and presents the most important data about the role of the videolaryngoendoscopic and stroboscopic evaluation of vocal fold leukoplakia.
Generally, the quality of the article is good, and the manuscript is interesting for the readers.
Overall, I consider the article could be a useful contribution.
Author Response
Reviewer 2
Dear Reviewer,
We would like to thank you very much for your effort and time devoted to reviewing our manuscript and for your kind words and positive feedback on our research.
Reviewer 3 Report
Thank you for the opportunity to review this paper. This is a well-executed project investigating the clinical use and efficacy of laryngoendoscopy and laryngostroboscopy in identifying the high-risk lekoplakia lesions. Strengths: statistics, figures, scientific questions. I strongly recommend this paper for publication if the following minor remarks are met:
11. Abstract – please state already here in the “conclusions” that the findings of this project can be used to guide the decision regarding immediate biopsy or watchful waiting.
22. Introduction – I find the introduction too long. Consider removing some parts or moving them to the Discussion. For example, line 45-48, remove?
33. Line 52 – what does the abbreviation LD stand for?
44. Line 53-54 – rewrite? A verb missing.
55. Line 78 – rewrite. Should it be “resulted in the improved resolution”?
66. Line 95 – LVS? Check spelling.
77. Material and methods – Line 106. Start with stating the number of participants and then report the number of patients with multiple lesions. Otherwise, it is confusing.
88. Smoking and alcohol consumption. How was it defined? How alcohol consumption was defined. Does the answer “No” means that the patient is a teetotaller?
99. State clearly inclusion and exclusion criteria. You listed only inclusion criteria in Materials and Methods.
110. Line 234 - change “was” to “were”
111. Line 288-289 – rewrite. Difficult to understand.
112. Line 362 – do you mean an additional diagnosis by biopsy?
113. Line 366 – which original study? By Chen et al.?
114. Overall the discussion is good. Please discuss the limitations of this study and how your study's findings can be used in the clinical practice. Do you plan to launch a website, app based tool where you could use your formula online? Or how do you want other clinicians to use your prediction model?
Author Response
Reviewer 3
Dear Reviewer,
Thank you very much for your effort and time taken to review our manuscript. We agree with all your comments and suggestions and we have implemented all the suggested changes in the manuscript, below is the point-by-point reply to your comments:
- Abstract – please state already here in the “conclusions” that the findings of this project can be used to guide the decision regarding immediate biopsy or watchful waiting.
Thank you for this valuable remark. We have added this information in the abstract – results as suggested
The findings of this study can be used to guide the decision regarding immediate biopsy or watchful waiting.
- Introduction – I find the introduction too long. Consider removing some parts or moving them to the Discussion. For example, line 45-48, remove?
We believe that the introduction is an important section that provides the readers with essential background information, and explains basic terminology and concepts. Therefore, we would like to keep this part of the manuscript unabridged.
- Line 52 – what does the abbreviation LD stand for?
LD stands for laryngeal dysplasia. We have clarified it in the manuscript.
- Line 53-54 – rewrite? A verb missing.
The missing verb has been added in the manuscript:
The underlying etiologies for the development of leukoplakia patches or plaques are long-term smoking and alcohol abuse, inhaled irritant substances, and viral infection such as HPV.
- Line 78 – rewrite. Should it be “resulted in the improved resolution”?
Thank you for this comment, indeed the word „improved” was missing. It has now been corrected in the manuscript:
The last two decades have seen a major technological advancement in endoscopic methods, which resulted in the improved resolution of images, contrast enhancement, magnification, and special filters for better exposure of the vascular structure of the mucosa.
- Line 95 – LVS? Check spelling.
It is a misspelling. The correct abbreviation is VLS, it has been corrected in the manuscript.
- Material and methods – Line 106. Start with stating the number of participants and then report the number of patients with multiple lesions. Otherwise, it is confusing.
The Patients paragraph has been rewritten according to the suggestion. Information on the exclusion criteria has been added.
- Smoking and alcohol consumption. How was it defined? How alcohol consumption was defined. Does the answer “No” means that the patient is a teetotaller?
Smoking was defined as:
- current or former smoker
- non-smoker
The idea was to determine whether the patients have been exposed to this risk factor.
With regard to alcohol consumption, the patients chose one of the following options:
- No (teetotaller)
- Occasional alcohol consumption
- Heavy/excessive alcohol consumption
None of the patients in the study group selected the heavy/excessive alcohol consumption option.
- State clearly inclusion and exclusion criteria. You listed only inclusion criteria in Materials and Methods.
The exclusion criteria were added to the manuscript:
The exclusion criterion was the disqualification of the patient from direct laryngoscopy with the collection of material for histopathological examination, e.g. when conservative treatment or work with a speech therapist could reduce pathological changes in the larynx. Patients who had previously undergone surgical interventions in the vocal folds or were diagnosed with leukoplakia in a location other than the glottis were not included in the study. Patients who cannot be operated on due to severe systemic diseases were excluded as well.
- Line 234 - change “was” to “were”
Corrected in the manuscript.
- Line 288-289 – rewrite. Difficult to understand.
The sentence was shortened to make it easier to understand:
Vocal cord leukoplakia is a clinical diagnosis regarded as a precancerous condition.
- Line 362 – do you mean an additional diagnosis by biopsy?
Thank you for this remark, the sentence was confusing. It has been clarified in the manuscript:
(…) whereas in cases of suspicious lesions (Type III) surgical treatment with histopathological verification) is required and it should not be delayed.
- Line 366 – which original study? By Chen et al.?
That’s correct. This information has been added to the manuscript:
In the original study by Chen et al [23],
- Overall the discussion is good. Please discuss the limitations of this study and how your study's findings can be used in clinical practice. Do you plan to launch a website, app based tool where you could use your formula online? Or how do you want other clinicians to use your prediction model?
The Limitations paragraph has been added to the manuscript:
The aim of the presented research was to present a ready-to-use, effective tool in daily clinical practice. In the future this could be extended with biological endoscopy methods. However, it should be borne in mind that they require a long learning curve and equipment availability.
In order to make the presented prediction algorithm readily accessible to other clinicians, we think about creating an application-based tool.
